# *CurrMask*: LEARNING VERSATILE SKILLS WITH AUTOMATIC MASKING CURRICULA

## ABSTRACT

Recent research in reinforcement learning (RL) has shown a growing trend towards the *pretraining* paradigm, where a unified model pretrained on diverse and unlabeled data can be quickly adapted to various downstream tasks. Inspired by advances in other domains, masked prediction provides a generic abstraction for pretraining on decision-making data by masking part of the trajectory and predicting the missing inputs. In spite of the versatility of masked prediction, it remains unclear how to balance the learning of reusable skills at different levels of complexity. To this end, we propose CurrMask, a curriculum masking approach that adjusts its masking scheme for learning diverse and versatile skills. The main idea behind CurrMask is that using masking schemes with different block sizes and mask ratios creates varying levels of temporal granularity. By explicitly combining them in a meaningful order, CurrMask can better capture both local dynamics and global dependencies. To achieve this, CurrMask uses a multi-armed bandit algorithm to find a proper curriculum for masking schemes that maximizes overall learning progress during training. Through extensive experiments, we show that CurrMask exhibits superior finetuning performance on offline RL tasks and zero-shot performance on goal-conditioned planning and skill prompting tasks. Additionally, our analysis reveals that CurrMask gradually increases the complexity of masking scheme, encouraging the model to capture both short-term and long-term dependencies.

## 1 INTRODUCTION

Humans distinguish themselves from machines by their capacity to adapt and generalize. One crucial factor behind this discrepancy is the drive to acquire reusable knowledge (e.g., concepts and behaviors) even in the absence of explicit reward (White, 1959). This has motivated research in unsupervised reinforcement learning (RL) (Laskin et al., 2021; Chebotar et al., 2021), in which the agent is required to learn from reward-free offline data (Carroll et al., 2022; Schwarzer et al., 2021) or online interaction (Liu & Abbeel, 2021; Yarats et al., 2021) for pretraining.

To build generic decision-making agents, great efforts have been made recently to apply self-supervised learning objectives for unsupervised offline pretraining (Schwarzer et al., 2021; Sun et al., 2023). Among these studies, one popular approach is *masked prediction*, a simple but versatile self-supervision framework that has proven its effectiveness in domains like language (Devlin et al., 2019) and vision (He et al., 2022). By masking a portion of the input trajectory and predicting it conditioned on the remaining, the model can not only capture rich representations but also learn transferable behaviors. For example, given a masked trajectory $(s_1, \texttt{[MASK]}, s_2, a_2, \texttt{[MASK]}, a_3)$, a model learned by masked prediction is forced to reason about both dynamics (i.e., masked state $s_3$) and behaviors (i.e., masked action $a_1$).

If masked prediction is the answer, what is the question? In this work, we investigate *which parts should be masked for decision-making data*. Our research stems from the finding that models trained with token-wise random masking, a widely adopted masking strategy in natural language modeling, fall short in modeling long-term dependencies (see Figure 4). While randomly masked words describe semantically dense information, state-action sequences in decision-making data contain heavy information redundancy (e.g., consecutive states are usually similar). This causes the model to predict masked tokens solely based on their neighboring unmasked tokens. Besides, state-action sequences naturally come with a unique pattern of interleaved modality, unlike single-modal word

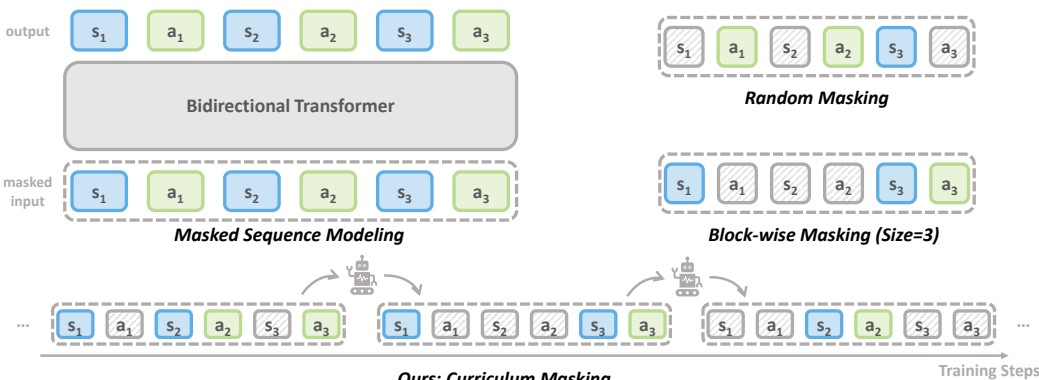

Figure 1: **Illustration of CurrMask.** Based on the framework of masked prediction, CurrMask incorporates masking schemes in various levels of temporal granularity and complexity to facilitate skill learning. To schedule different masking schemes in a meaningful order, CurrMask leverages a multi-armed bandit model to adjust its mask based on the learning progress during training.

sequences. These discrepancies in *information density* and *sequence pattern* requires a reevaluation of masking scheme design for RL pretraining.

Motivated by the above considerations, we posit that masking for decision-making data needs to be done in blocks rather than tokens. A block of consecutive state and action tokens forms a semantic entity of *skill* (Ajay et al., 2021; Pertsch et al., 2021). For example, applying block-wise masking of size 3 results in $(s_1, [\texttt{MASK}], [\texttt{MASK}], [\texttt{MASK}], s_3, a_3)$. By using a block-wise masking scheme, the model is compelled to prioritize global dependencies over basic local correlations when making masked predictions. Moreover, the combination of multiple mask ratios and block sizes incentivizes the model to acquire adaptable skills that can be utilized for diverse downstream tasks.

To encourage the model to capture semantically meaningful relations, we propose CurrMask, a curriculum masked prediction approach that automatically arranges the order of different masking schemes for training. Our main intuition is that the ability of long-horizon reasoning can be developed by first learning how to act locally. This motivates us to consider a mask curriculum representing a skill learning curriculum. We use the EXP3 algorithm (Auer et al., 2002) to determine which masking scheme to apply when faced with uncertainty of training dynamics, in order to establish a meaningful order. An illustration of our approach is shown in Figure 1.

We conduct a series of empirical studies on various MuJoCo-based control tasks, including locomotion and robotic arm manipulation. Our results demonstrate that CurrMask enables the learning of a versatile model that achieves superior performance in representation learning, zero-shot skill prompting, and zero-shot goal-conditioned planning. Further analysis reveals that CurrMask effectively mitigates the issue of local correlations and is better in capturing long-term dependencies. These findings shed light on the design of masking schemes that can effectively balance the acquisition of reusable skills at varying levels of temporal granularity and complexity.

## 2 RELATED WORK

Our work considers **masked prediction** as an effective approach for **unsupervised RL pretraining**. By incorporating **curriculum learning** into the framework, we show that the proposed approach CurrMask can effectively boost masked prediction. In this section, we give a review over related works in these research directions.

**Masked Prediction as a Self-Supervision Task**   Masked prediction requires the model to predict a missing portion of the input that has been held out. Pretraining via masked prediction has been explored in natural language processing (Devlin et al., 2019; Joshi et al., 2020), computer vision (He et al., 2022; Bao et al., 2022), and decision making (Liu et al., 2022; Carroll et al., 2022). One of the most notable applications is masked language modeling (Devlin et al., 2019) for learning transferable

text representations. Recently, it has been shown that masked prediction can also facilitate decision making, either by training visual backbones (Radosavovic et al., 2023; Seo et al., 2023) or by learning temporal information (Carroll et al., 2022; Liu et al., 2022).

**Masking Schemes**   For masked prediction, a central question is *what is masked*. One common scheme is to randomly mask some of the tokens from the input. Apart from random masking, recent studies have proposed attention-guided masks (Li et al., 2021; Kakogeorgiou et al., 2022; Li et al., 2022) and adversarial masks (Shi et al., 2022; Tomar et al., 2023) to force the model to focus specific parts of the input for better performance. In the domain of decision making, previous work usually considers the simplest random masking scheme (Liu et al., 2022), manually designed task-specific masks (Carroll et al., 2022), or a combination of random masking with other representation learning objectives (Sun et al., 2023). In this work, we aim to illustrate the connection between masking schemes and skill learning, in search for automatic learning curricula for masked prediction.

**Unsupervised RL Pretraining**   Our work also falls into the category of extracting prior knowledge without extrinsic human supervision for sample-efficient RL. Previous work has vastly studied reward-free RL, in which the agent can interact with the environment in the absence of rewards (Eysenbach et al., 2019; Yarats et al., 2021). Another setting is to utilize unlabeled offline data for representation learning (Schwarzer et al., 2021; Stooke et al., 2021) or skill learning (Ajay et al., 2021; Jiang et al., 2022). Masked prediction presents a promising framework to enjoy the best of both world.

**Curriculum Learning**   Inspired by how humans learn faster when knowledge is ordered by easiness, curriculum learning (Elman, 1993; Bengio et al., 2009) has been formulated for machine learning algorithms to improve training efficiency. While curriculum learning has been actively explored in the context of online RL (Jabri et al., 2019; Fang et al., 2021), in this work we show that offline RL pretraining also benefits from a proper learning curriculum.

## 3   PRELIMINARIES

### 3.1   MASKED PREDICTION

Let $\tau = (s_t, a_t)_{t=1}^{T} = (s_1, a_1, s_2, a_2, \cdots, s_T, a_T)$ denote a trajectory consisting of state-action sequences and $\mathcal{D}$ denote the training dataset. The self-supervised task of masked prediction is to reconstruct $\tau$ from a masked view $\mathtt{masked}(\tau)$, where $\mathtt{masked}(\cdot)$ represents a specific masking function. For example, if $\mathtt{masked}(\cdot)$ represents a deterministic scheme that masks the initial and final actions of the input, the resulting masked trajectory is $\mathtt{masked}(\tau) = (s_1, [\mathtt{MASK}], s_2, a_2, \cdots, s_T, [\mathtt{MASK}])$. Here, $[\mathtt{MASK}]$ represents a special learnable token. The learning objective[1] is then given by:

$$\max_{\theta} \mathbb{E}_{\tau \sim \mathcal{D}} \sum_{t=1}^{T} \log P_{\theta} \left( s_t, a_t \mid \mathtt{masked}(\tau) \right),$$

where $P_{\theta}$ is parameterized by a bidirectional transformer (Devlin et al., 2019). By reconstructing state-action sequences, the model learns to reason over temporal dependencies.

Importantly, the choice of $\mathtt{masked}(\cdot)$ specifies a concrete task the model is trained on. Therefore, it is crucial to design an appropriate masking scheme that enables learning of general relationships in state-action sequences. This goal boils down to two aspects: 1) *how much is masked*, and 2) *what is masked*. For the former, it has been shown that a high mask ratio (e.g., 95%) is meaningful for decision-making data due to its low information density (Liu et al., 2022). For the latter, since it is undesirable to specify the tasks of interest when pretraining, the random masking scheme is widely used (i.e., uniformly sampling a subset of tokens to mask). These principles form the basis of our proposed masking approach, which is elaborated in Section 4.

---

[1]It is a design choice to compute loss on the entire input (Vincent et al., 2008) or only on the masked tokens (Devlin et al., 2019). In this work, we apply the former as it has been shown to work better with sequential decision-making data (Liu et al., 2022).

---

**Algorithm 1:** Curriculum Masking

---

**Input :** candidate masking schemes $\mathcal{M}$ with cardinality $K$, training steps $T$, evaluation interval $I$, evaluation samples $N$, offline dataset $\mathcal{D}$, bidirectional transformer $P_\theta$

1 Initialize weights $w_k \leftarrow 0$ for $k = 1, \ldots, K$;
2 Initialize masking scheme $k \sim \pi_{\mathbf{w}}$;
3 **for** $t \leftarrow 1 \ldots T$ **do**
   /* Evaluation & selection                                          */
4    **if** $t \mod I = 0$ **then**
5      Compute loss $\mathcal{L}_{\text{target}}$ on $\{\tau_n \mid \tau_n \sim \mathcal{D}, n = 1, \ldots, N\}$;
6      Calculate reward $r$ (Equation 1);
7      Update weights $w_k \leftarrow 0$ for $k = 1, \ldots, K$ (Equation 2);
8      Update masking scheme $k \sim \pi_{\mathbf{w}}$;
9    **end**
   /* Training                                                        */
10    Compute loss $\mathcal{L}_k$ on $\tau \sim \mathcal{D}$;
11    Update $\theta$ by gradient descent;
12 **end**

---

## 3.2 AUTOMATED CURRICULUM LEARNING

Automated curriculum learning considers how to arrange the order of tasks during training by adapting the selection of learning scenarios to match the learner's abilities. Consider a series of tasks represented by loss functions $\mathcal{L}_1, \ldots, \mathcal{L}_K$. The objective is to find a time-varying sequence of tasks to accelerate training. To this end, a proper automatic curriculum needs to specify two factors: 1) how to measure *learning progress*, in order to adjust its task schedule dynamically, and 2) how to perform *task selection* based on progress signals. We describe our design in Section 4.

## 4 CURRICULUM MASKED PREDICTION

In this section, we describe the proposed approach, CurrMask, for unsupervised RL pretraining. Algorithm 1 summarizes the overall pipeline. At the core of CurrMask is masked prediction as a versatile self-supervised learning objective and an automatic learning curriculum over masking schemes to enable fast skill discovery. Once pretrained on offline data, CurrMask can perform various downstream tasks in a zero-shot manner, or be finetuned for policy learning. In the following, we elaborate the design of CurrMask and provide sufficient explanation.

### 4.1 BLOCK-WISE MASKING ENHANCES LONG-TERM REASONING

Our research is based on the discovery that models trained using random masking, a commonly used strategy in natural language modeling, fall short in capturing long-term dependencies (see Figure 4). This is undesirable for decision-making agents that maximize long-term reward. To overcome this issue, CurrMask applies the block-wise masking scheme (Joshi et al., 2020; Bao et al., 2022) that masks the trajectory in blocks instead of individual tokens. By doing so, CurrMask pushes the model to focus on semantically meaningful abstractions rather than simple local correlations. Predicting missing blocks of state-action sequences also resembles multi-step inverse dynamics models (Lamb et al., 2022), which has been shown to learn robust representations for decision making. We present pseudocode of our block-wise masking implementation in Appendix A.

Blocks consisting of consecutive states and actions also form a notion of *skills* or *primitives*. Prior works in offline skill discovery (Ajay et al., 2021; Jiang et al., 2022) typically use variational inference to partition trajectories into skills. In this work, we argue that masked prediction with block-wise masking represents an alternative approach for offline skill discovery. The link between masked prediction and skill discovery inspires us to explore automatic curricula that can aid in learning skills.

## 4.2 LEARNING OVER A MIXTURE OF MASKING SCHEMES

The block size explicitly determines the level of temporal granularity for masked prediction. To capture both short-term and long-term temporal dependencies, CurrMask employs a combination of masking schemes with varying block sizes and mask ratios during pretraining.

Given a set of masking schemes $\mathcal{M}$ where $|\mathcal{M}| = K$, we define the loss function for masked prediction task $k$ as:

$$\mathcal{L}_k(\tau;\theta) = \sum_{t=1}^{T} \log P_\theta\left(s_t, a_t \mid \texttt{masked}_k(\tau)\right),$$

where $\texttt{masked}_k \in \mathcal{M}$ denotes a specific masking scheme. CurrMask aims to minimize the multi-task learning objective $\mathcal{L}_{\text{target}}(\tau;\theta) = \frac{1}{K}\sum_{k=1}^{K}\mathcal{L}_k(\tau;\theta)$.

## 4.3 AUTOMATED CURRICULUM LEARNING BOOSTS TRAINING EFFICIENCY

A key feature of mixed masking schemes is their inherent variability in complexity. Intuitively, the ability of reasoning over global dependencies can be developed by first learning how to plan within a short horizon. This motivates us to consider curriculum learning to facilitate masked prediction. By scheduling masking schemes in a meaningful order, we expect that the model will learn more efficiently and quickly during training.

**Evaluation of Learning Progress**   The first factor to determine is the measure of learning progress. Ideally, we would like the curriculum to maximize the rate at which the model learns to solve downstream tasks. However, it is usually intractable to measure without downstream task information. Hence, we consider *target loss decrease* (Graves et al., 2017) as a proxy signal for learning progress:

$$r = f_{\text{scale}}(\mathcal{L}_{\text{target}}(\theta) - \mathcal{L}_{\text{target}}(\theta')), \tag{1}$$

where $\theta$ and $\theta'$ denote the model parameters before and after training, respectively. To alleviate the issue of time-varying magnitudes, we follow Graves et al. (2017) to rescale values into $[-1, 1]$ using the 20-th and 80-th percentiles of history values: $f_{\text{scale}}(\hat{r}) = \max(-1, \min(1, \frac{2(\hat{r}-r^{\text{lo}})}{r^{\text{hi}}-r^{\text{lo}}} - 1))$.

**Task Selection**   To schedule tasks in a meaningful order, we aim to minimize $\mathcal{L}_{\text{target}}$ achieved after training on them sequentially. This can be formulated a multi-armed bandit problem (Lattimore & Szepesvári, 2020), where each arm represents a masking scheme and the goal is to maximize the total reward earned over time. Since the reward distribution induced by Equation 1 shifts as the network learns, we use the EXP3 algorithm (Auer et al., 2002), a non-stochastic multi-armed bandit algorithm that mixes the probability distribution computed using exponential weights $\mathbf{w}$ with the uniform distribution:

$$\pi_{\mathbf{w}}(i) = (1 - \gamma)\frac{w_i}{\sum_{j=1}^{K} w_j} + \frac{\gamma}{K} \quad i = 1, \dots, K.$$

Each time EXP3 samples an arm $k \sim \pi_{\mathbf{w}}$ and observes reward $r$, it uses the importance-weighted estimator $\hat{x}_i = \frac{\mathbb{I}\{i=k\}r}{\pi_{\mathbf{w}}(i)}$ to update its weights according to the following formula:

$$w'_i = w_i \exp\left(\gamma\hat{x}_i/K\right) \quad i = 1, \dots, K. \tag{2}$$

As such, exponential growth significantly increases the probability of choosing good arms (i.e., masking schemes). Please see Appendix D for discussions on how CurrMask addresses non-stationarity.

## 5 EXPERIMENTS

In this section we conduct an empirical study to answer the following questions: **(Q1)** Can CurrMask learn a versatile model that achieves good performance on a variety of downstream tasks, both in zero-shot and finetuning scenarios? **(Q2)** What role do block-wise masking and masking curricula play in CurrMask? **(Q3)** Does CurrMask better capture long-term temporal dependencies?

| reward ↑ | run | stand | walk | reach-bl | reach-br | reach-tl | reach-tr |
|---|---|---|---|---|---|---|---|
| Random | $30.4_{\pm 0.9}$ | $105.1_{\pm 3.4}$ | $58.7_{\pm 4.0}$ | $64.3_{\pm 5.5}$ | $63.1_{\pm 4.2}$ | $62.4_{\pm 4.3}$ | $67.4_{\pm 3.9}$ |
| Mixed | $30.6_{\pm 1.4}$ | $110.4_{\pm 3.3}$ | $50.9_{\pm 2.2}$ | $70.4_{\pm 7.6}$ | $64.8_{\pm 3.7}$ | $65.8_{\pm 3.9}$ | $68.9_{\pm 5.3}$ |
| Mixed-inv | $24.6_{\pm 1.3}$ | $105.5_{\pm 5.0}$ | $55.3_{\pm 3.1}$ | $59.8_{\pm 3.3}$ | $57.2_{\pm 2.7}$ | $62.8_{\pm 2.1}$ | $62.7_{\pm 2.8}$ |
| Mixed-prog | $27.0_{\pm 0.5}$ | $105.2_{\pm 3.7}$ | $52.8_{\pm 4.8}$ | $86.8_{\pm 6.4}$ | $87.7_{\pm 5.1}$ | $87.9_{\pm 4.8}$ | $89.2_{\pm 3.8}$ |
| CurrMask | $37.1_{\pm 1.4}$ | $109.7_{\pm 3.3}$ | $90.6_{\pm 2.4}$ | $81.7_{\pm 7.4}$ | $84.5_{\pm 2.9}$ | $78.7_{\pm 6.1}$ | $83.5_{\pm 3.7}$ |

Table 1: **Skill prompting results.** We report the zero-shot performance of models pretrained with different masking schemes. Results are averaged over 5 random seeds.

## 5.1 ENVIRONMENT SETUP

We evaluate our method on a set of environments from the DeepMind control suite (Tunyasuvunakool et al., 2020). Each environment has several tasks specified by how the reward function is defined. Specifically, we consider a total of 7 tasks that are associated with 2 different environments. At evaluation, we test how well the model pretrained for each environment on offline datasets adapts to different downstream tasks. For more details and experimental results, please refer to Appendix B.2 and Appendix E, respectively.

**Environments** The `walker` environment consists of 3 locomotion tasks (`run`, `stand`, and `walk`). All the tasks provide a dense reward measure of task completion. For example, task `run` provides rewards encouraging forward velocity. `jaco` is an environment for robot arm manipulation, which includes 4 reaching tasks (`bottom_left`, `bottom_right`, `top_left`, and `top_right`). These tasks are sparse-reward tasks given that nonzero rewards are provided only when the current position is within a certain distance threshold of the target position.

**Dataset Collection** For each environment, we construct a multi-task dataset by collecting trajectories of 12M steps from the replay buffer of TD3 agents (Fujimoto et al., 2018). Specifically, for each task in the Walker/Jaco environment, we train an agent for 4M/3M environment steps. This collection procedure ensures that the pretraining dataset contains experiences of varying quality. For zero-shot evaluation, we additionally construct a validation set for each environment using the same protocol but with different random seeds.

**Baselines & Implementation Details** We compare CurrMask with the following baselines: 1) `Random` (Liu et al., 2022) samples a mask ratio and randomly masks a portion of individual tokens in each training step; 2) `Mixed` samples a mask ratio as well as a block size at each step and applies block-wise masking; 3) `Mixed-prog` uses a manually designed mask curriculum that progressively increases the block size during pretraining; 4) `Mixed-inv` reverses the order of `Mixed-prog`'s mask curriculum. We consider multiple mask ratios (15%, 35%, 55%, 75%, and 95%) and multiple block sizes $(1, 2, \ldots, 20)$ to construct scheme pool $\mathcal{M}$ $(|\mathcal{M}| = 100)$ for CurrMask. The baselines are subject to the same candidate ratios and block sizes. For all the evaluated methods, we use the same encoder-decoder transformer architecture with a 3-layer encoder and a 2-layer decoder, following prior work (He et al., 2022; Liu et al., 2022). The encoder takes only unmasked states and actions into account, whereas the whole trajectory including both masked and unmasked tokens will be passed to the decoder. Both the encoder and decoder are bidirectional, i.e., each token can attend to both the left and right context via the self-attention mechanism.

## 5.2 DOWNSTREAM TASKS

To demonstrate the versatility of CurrMask, we consider various downstream tasks that require different capabilities, including zero-shot inference by specifying certain masking schemes (i.e., skill prompting and goal-conditioned planning) and adaptation via finetuning (i.e., offline RL).

**Skill Prompting** A unified model trained on diverse multi-task data is expected to acquire various skills that can be invoked to perform certain tasks. Skill prompting tests this ability by requiring the model to generate consecutive behaviors given a short state-action sequence. For each task, we randomly sample 5-timestep state-action sequences from the validation set as the prompts, and evaluate the quality of the generated trajectory of length 120 by its task rewards.

| distance $\downarrow$ | run | stand | walk | reach-bl | reach-br | reach-tl | reach-tr |
|---|---|---|---|---|---|---|---|
| Random | $15.45_{\pm.47}$ | $4.83_{\pm.44}$ | $10.11_{\pm.21}$ | $1.50_{\pm.06}$ | $1.51_{\pm.04}$ | $1.45_{\pm.05}$ | $1.44_{\pm.06}$ |
| Mixed | $15.35_{\pm.51}$ | $4.80_{\pm.43}$ | $10.16_{\pm.24}$ | $1.55_{\pm.07}$ | $1.57_{\pm.05}$ | $1.53_{\pm.05}$ | $1.52_{\pm.07}$ |
| Mixed-inv | $16.53_{\pm.57}$ | $5.28_{\pm.49}$ | $11.18_{\pm.29}$ | $1.67_{\pm.07}$ | $1.63_{\pm.06}$ | $1.66_{\pm.07}$ | $1.60_{\pm.07}$ |
| Mixed-prog | $15.66_{\pm.55}$ | $4.95_{\pm.47}$ | $10.19_{\pm.20}$ | $1.56_{\pm.05}$ | $1.55_{\pm.04}$ | $1.53_{\pm.05}$ | $1.51_{\pm.09}$ |
| CurrMask | $15.48_{\pm.49}$ | $4.80_{\pm.44}$ | $10.05_{\pm.20}$ | $1.45_{\pm.05}$ | $1.47_{\pm.05}$ | $1.41_{\pm.06}$ | $1.38_{\pm.08}$ |

Table 2: **Goal-conditioned planning results.** We report the zero-shot performance of models pretrained with different masking schemes. Results are averaged over 5 random seeds.

**Goal-conditioned Planning**    Another type of downstream task we consider is goal-conditioned planning. Starting from a given state, the model needs to roll out actions that can achieve target goals within a number of steps. We consider 5 target goals with intervals of 20 steps, seeking to evaluate the model's capability to generate long-term plans. For each task, we use a fix set of trajectories sampled from the validation set. We randomly sample starting states and corresponding goal states from the trajectories for planning. The performance is assessed by the L2 distance between every goal and the closest state within the short range of a given time budget.

**Offline RL**    Finally, we study if the representations learned by CurrMask can accelerate offline RL. Our intuition is that, by capturing both short-term and long-term temporal dependencies in the offline data, CurrMask should learn useful and generic representations for downstream finetuning. For each task, we add a critic head and actor head on top of the encoder, and run TD3 (Fujimoto et al., 2018) to perform offline RL training[2], following prior work (Liu et al., 2022). Each offline dataset is collected from the entire replay buffer of a ProtoRL agent (Yarats et al., 2021) trained for 2M environment steps. Notably, the datasets consist of highly exploratory data, which emphasizes the importance of having good representations.

## 5.3    MAIN RESULTS

We test the versatility of CurrMask over a variety of downstream tasks, in answer to **Q1** and **Q2**.

**Skill Prompting**    Table 1 summarizes the zero-shot performance for skill prompting. We can observe that CurrMask consistently achieves better performance than `Random`, suggesting that CurrMask is proficient in mirroring skill prompts to accomplish specific tasks. Besides, other baseline methods that incorporate block-wise masking (i.e., `Mixed`, `Mixed-prog`, and `Mixed-inv`) generally outperform `Random`. This matches our expectation that blocks form more semantically meaningful entities than individual tokens and can be utilized by masked prediction to facilitate skill learning. The only exception is `Mixed-inv`. The poor performance of `Mixed-inv` sends a strong signal that a proper curriculum is important for masked prediction training.

**Goal-conditioned Planning**    Next we evaluate how capable CurrMask is for long-horizon planning. As shown in Table 2, CurrMask can rollout better goal-reaching trajectories than the baselines in 6 out of 7 tasks. The advantages are more prominent in the `jaco` environment. We conjecture that it is because robot arm manipulation forms a more hierarchical structure to reach a specific goal. Another notable observation is that, in contrast to skill prompting results, `Mixed` has worse performance than `Random`. This indicates that the superior performance of CurrMask is not only due to block-wise masking, but rather a consequence of dynamically balancing what to mask during training.

**Offline RL**    Finally, we present offline RL results in Figure 2. Compared with learning from scratch, learning with pretrained representations obtained by CurrMask results in significant training speedup and performance improvement. We observe that CurrMask outperforms `Random`, which suggests that CurrMask not only learns diverse skills but also extracts transferable representations for policy learning. It should also be noted that in some cases (e.g., `walk`) pretraining with `Random` leads to diminished performance for finetuning, whereas CurrMask is generally more stable.

---

[2]Although TD3 is originally designed as an off-policy RL algorithm, Yarats et al. (2022) show that it achieves very competitive performance on offline datasets of diverse behaviors.

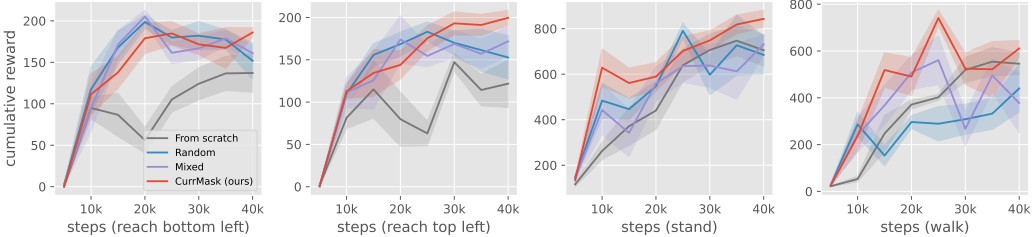

Figure 2: **Offline RL results.** We report the finetuning performance of models pretrained with different masking schemes. Results are averaged over 5 random seeds.

## 5.4 Analysis

In this section, we investigate several aspects of CurrMask to further answer **Q2** and **Q3**.

**Impact of Block-wise Masking** To better understand how block-wise masking contributes to CurrMask, we conduct an ablation study on the choice of block sizes. Figure 3a shows the influence of the block size, where masked prediction is combined with randomly selected mask ratios and a fixed block size. With block-wise masking, masked prediction benefits from larger block sizes to perform zero-shot skill prompting. Besides, mixing different block sizes uniformly for training, referred to as `Mixed`, leads to average performance. This indicates that the block size does have a great impact over final performance, and a proper learning curriculum can be crucial.

**Impact of Masking Curricula** Another important question is whether good masking curricula should be determined manually or found adaptively during training. We would like to emphasize that `Mixed-prog` does not consistently lead to performance improvements compared to `Random`. Specifically, in most goal-conditioned planning tasks and in offline RL and skill prompting tasks of the walker domain, Mixed-prog performs worse than `Random`. The performance gap can be very substantial (e.g., a decrease of 56.8% in offline RL performance for `walk`). In contrast, for CurrMask, we consistently observe improvements over `Random`.

We provide additional experimental results to better illustrate that CurrMask is not just rediscovering the programmatic curriculum. Figure 6 displays the skill prompting results versus training steps during pretraining, revealing noticeable differences in skill learning progress between `Mixed-prog` and CurrMask. Additionally, Table 6 compares CurrMask with `Mixed` using a constant block size. This comparison emphasizes that a proper masking scheme is unlikely to be predetermined and highlights the benefits of CurrMask for its adaptivity.

**Training Dynamics** Next, we investigate how automatic curricula steer masking schemes during training. Figure 3b visualizes the time-varying probabilities of choosing different block sizes

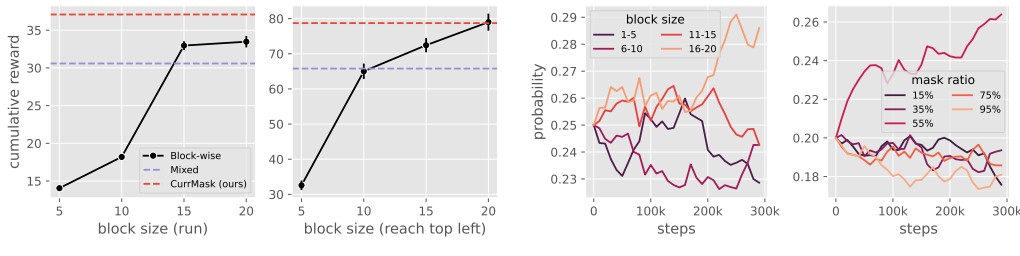

(a) Impact of block-wise masking      (b) Impact of masking curricula (reach top left)

Figure 3: **Both block-wise masking and curriculum masking contribute to CurrMask's performance.** Left: the performance of zero-shot skill prompting as a function of fixed block size. Right: the probabilities of choosing different block sizes and mask ratios during pretraining with CurrMask.

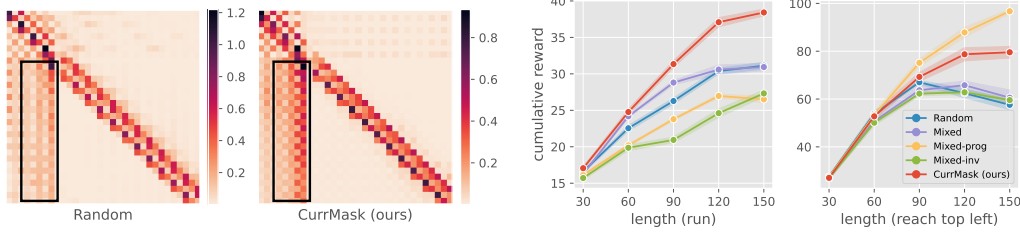

(a) Attention maps (reach top left)

(b) Skill prompting performance vs. rollout length

Figure 4: **Analysis of long-term prediction capability.** Left: We visualize the attention map (L2-normalized over different heads) of the first decoder layer, when the model is conducting skill prompting. Right: the performance of zero-shot skill prompting as a function of rollout length.

and mask ratios during CurrMask pretraining. We can see that CurrMask gradually increases the probability of choosing large block sizes while also preferring a moderate mask ratio. The former observation reveals that CurrMask has a tendency to learn more complex skills, which aligns with our intuition. For the latter, we believe it reflects the degree of information redundancy in sequential decision-making data, also reported in previous work (Liu et al., 2022; He et al., 2022).

**Evaluation of Long-term Prediction** One of the most important intuition behind CurrMask is that block-wise masking can enhance the model's capability to capture long-term dependencies. To verify this, we look into the attention maps during prediction with skill prompts even when they are far from current timesteps. Figure 4a compares the attention patterns induced by `Random` and CurrMask. We notice significant differences in how the approaches use the prompt. Compared to `Random`, CurrMask better leverages the provided states and actions to generate behaviors. Besides, the predictions of CurrMask attend to prior actions more than they do to prior states. These findings support our intuition that CurrMask is more effective at extracting useful long-term dependencies. We offer a more comprehensive assessment in Appendix F.

This discrepancy in attention patterns is further validated by the performance of long-horizon skill prompting. Figure 4b shows the skill prompting performance as a function of the rollout length. We observe that CurrMask outperforms the baselines significantly when the rollout length is extended. Notably, `Random` and `Mixed-inv` have degenerated performance for long rollouts, supporting our hypothesis that CurrMask acquires non-trivial long-term prediction capacity.

## 6 CONCLUSION

In this work, we propose CurrMask, a curriculum masking approach for unsupervised RL pretraining. Motivated by the unique pattern of sequential decision-making data (i.e., *low information density* and *interleaved modality*), we propose to apply block-wise masking with mixed mask ratios and block sizes to capture temporal dependencies at both short-term and long-term levels of granularity. As different masking schemes naturally vary in prediction difficulty, we consider automated curriculum learning as the inner drive to facilitate training by scheduling these schemes in a meaningful order. We show through extensive experiments that CurrMask learns a versatile model that consistently outperforms the baselines in various downstream tasks. Our analysis of the impact of block-wise masking and curriculum learning emphasizes the adaptivity of CurrMask and its superior ability to extract global dependencies.

**Limitations** One limitation of CurrMask is the computational overhead, as CurrMask relies on an extra bandit model to schedule masking schemes for training[3]. Furthermore, the advantages offered by CurrMask could be affected by the underlying structure of the environment. This encourages us to extend our method to more challenging settings like image-based RL in future research.

---

[3]We observe a computation overhead of 4.7% for 100k gradient steps with a single RTX 3090 graphic card.

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

# A   PSEUDOCODE OF BLOCK-WISE MASKING

Algorithm 2 demonstrates the block-wise masking mechanism, which is employed as an intermediate step for masking in CurrMask and other baseline models.

---

**Algorithm 2:** Block-wise Masking

---

**Input** : sequence length $L$, mask ratio $p$, block size $b$
**Output :** binary mask matrix $m \in \{0,1\}^L$ (0 for masked, 1 for unmasked),

1  **if** $b = 1$ **then**
2  | **return** random_masking($L, p$);
3  **end**
4  $l \leftarrow p \cdot L$ ;                                     /* length of masked tokens */
5  $c \leftarrow \lfloor \frac{L-1}{b} \rfloor$ ;                 /* number of blocks in a sequence */
6  Initialize mask $m \leftarrow \mathbb{1}$;
7  Randomly choose start index $s \leftarrow$ random($0, \cdots, b-1$);
8  Shuffle block indices $bs \leftarrow$ shuffle($0, \ldots, c-1$);
9  Expand block indices to token indices $ts \leftarrow (i \cdot b + j + s$ for $i$ in $bs$ for $j$ in $(1, \cdots, b-1)$);
10  Mask tokens $m \leftarrow$ set_zero($m, ts[-l :]$);
11  **if** len($ts$) $< l$ **then**
12  | Mask remaining tokens $m \leftarrow$ set_zero($m, ((0, \cdots, L-1) - ts)[\text{len}(ts) - l :]$);
13  **end**
14  **return** $m$;

---

# B   EXPERIMENTAL DETAILS

## B.1   DATA COLLECTION

**Pretraining Datasets**    For both the walker and jaco environments, we create a multi-task dataset by gathering trajectories from the replay buffer of TD3 agents. We collect a total of 12M steps from the replay buffer for each environment. Each task in the walker environment is trained for 4M environment steps, while each task in the jaco environment is trained for 3M steps. By following this procedure, we ensure that the pretraining datasets encompasses experiences of varying quality.

**Validation Datasets**    For zero-shot evaluation of both skill prompting and goal-conditioned planning, we construct a separate validation set for each environment using the same collection protocol for pretraining datasets but with different random seeds.

**Training Datasets for Offline RL**    Each offline dataset is obtained from the complete replay buffer of a ProtoRL agent, which was trained for 2M environment steps. For each task, the collected dataset is relabeled with task-specific rewards during offline RL. It is worth mentioning that these datasets contain highly exploratory data, emphasizing the significance of having effective representations. Table 3 summarizes the statistics.

| task | min | max | mean |
|------|-----|-----|------|
| stand | 27.07 | 408.59 | 198.85 |
| walk | 4.81 | 199.95 | 72.95 |
| run | 4.55 | 79.00 | 38.61 |
| top_left | 0.00 | 245.18 | 5.23 |
| top_right | 0.00 | 227.04 | 5.64 |
| bottom_left | 0.00 | 225.56 | 3.59 |
| bottom_right | 0.00 | 242.03 | 4.35 |

Table 3: Episodic return statistics of training datasets used for offline RL.

### B.2 Implementation Details

**Hyperparameters**  Our CurrMask implementation is based on the MaskDP codebase[4]. Table 4 summarizes the hyperparameters used by CurrMask for training and evaluation.

| model | value |
|---|---|
| # encoder layers | 3 |
| # decoder layers | 2 |
| # attention heads | 4 |
| context length | 64 |
| hidden dimension | 256 |
| mask ratio | $[15\%, 35\%, 55\%, 75\%, 95\%]$ |
| block size | $[1, 2, \ldots, 20]$ |
| **training** | |
| optimizer | Adam |
| batch size | 384 |
| learning rate | 1e-4 |
| # gradient steps | 300k |
| EXP3 $\gamma$ | 0.2 |
| evaluation interval $I$ | 100 |
| # evaluation samples $N$ | 10 |
| **skill prompting** | |
| # seeds | 5 |
| # trajectories sampled per seed | 100 |
| prompt length | 5 |
| rollout length | 120 |
| **goal-conditioned planning** | |
| # seeds | 10 |
| # trajectories sampled per seed | 100 |
| **offline RL** | |
| # seeds | 5 |
| # training steps | 35k |

Table 4: Hyperparameters used for model training and evaluation.

**Baselines**  For all the baselines, we use the same model architecture and common hyperparameters as CurrMask. The implementation of `Mixed-prog` involves manually partitioning the training process into four stages based on the value of `current_step/total_step`. Within the intervals of $[[0, 0.25), [0.25, 0.5), [0.5, 0.75), [0.75, 1.00)]$, sub-sequences are sampled with lengths ranging from $[[1, 5], [1, 10], [1, 15], [1, 20]]$, respectively. This deliberate control enables the progressive increase in block size of the mask, posing greater challenges to the training procedure as it unfolds. The implementation of `Mixed-inv` shares significant similarities with `Mixed-prog`. Both methods adopt a four-stage approach to partition the training process. The key distinction lies in the sampling of sub-sequence lengths as training progresses. In the case of `Mixed-inv`, these lengths follow a descending pattern, specifically $[[1, 20], [1, 15], [1, 10], [1, 5]]$.

**Evaluation of Skill Prompting**  To facilitate skill prompting, the agent is provided with a short state-action segment randomly extracted from a trajectory in the validation dataset. The agent is then positioned at the final state of the segment and tasked with generating subsequent behaviors in an autoregressive manner. The quality of the generated sequence is evaluated by comparing its accumulated rewards with those obtained from the rollout of an expert with advanced skills. In detail, we employ a prompt length of 5 timesteps and the initial position of each prompt is randomly sampled within the range of $[0.1 \cdot \texttt{trajectory\_length}, 0.85 \cdot \texttt{trajectory\_length}]$. Therefore, the prompt may be located at the beginning of a trajectory or skewed towards the later stages, resulting in the agent's state being in a low-speed starting phase or a high-speed running phase in the cases of the `walk`/`run` task.

---

[4] `https://github.com/FangchenLiu/MaskDP_public`

**Evaluation of Goal-conditioned Planning**  To implement goal-conditioned planning, we randomly sample a goal context of length 100 from the trajectories in the validation set. The position of the goal is set at specific locations $[20, 40, 60, 80, 100]$. The agent is initially placed at the starting position of the goal context, and the rollout continues for the remaining tokens. We calculate the L2 distance between each goal state and its closest state token within the rollout length as a metric for evaluation.

**Evaluation of Offline RL**  In offline RL, the main objective is to train a model to maximize the return for a specific task, as defined by a reward function. This differs from our self-supervised pretraining objective, so additional finetuning is required. To align with the RL setting, we modify the bidirectional attention mask in the transformer to a causal attention mask. This change allows the model to attend only to previous states and actions during training, simulating the sequential nature of RL tasks. We also utilize a standard actor-critic framework similar to TD3 by incorporating a critic head and an actor head on top of the pretrained encoder. The actor takes a sequence of states as input, while the critic takes a sequence of state-action pairs as input. Both components operate without any masking. Then we perform RL training using the modified architecture.

## C  SKILL PROMPTING VISUALIZATION

To supplement the evaluation of skill prompting, we provide qualitative results of the generated actions as shown in Figure 5. One interesting finding is about skill prompting on the `run` task. Both agents trained with `Random` and CurrMask fall down in the initial steps, which could be due to the high velocity. As the trajectory is rolled out, the agent trained with `Random` struggles to stand up. On the other hand, the agent trained with CurrMask manages to stand up and continue running. This also supports our observation that CurrMask can better leverage information given by the prompt, even when conducting long-horizon prediction.

## D  DISCUSSIONS ON NON-STATIONARITY

Non-stationarity is a major challenge for algorithm design in our context. We want to emphasize two important properties: 1) The reward distribution is non-stationary, and 2) Despite this, the learning process usually progresses gradually without sudden regime shifts (Zhou et al., 2021).

For the former, we want to emphasize that our method tackles the non-stationarity in two aspects. Firstly, EXP3, a special case of online mirror descent, is inherently adaptable to reward distributions that change over time. Secondly, we alleviate this issue by rescaling rewards using historical percentiles. For the latter, while abrupt distribution changes are not typically observed, we believe that our framework can easily accommodate other techniques like sliding windows and reward discounting to address significant non-stationarity.

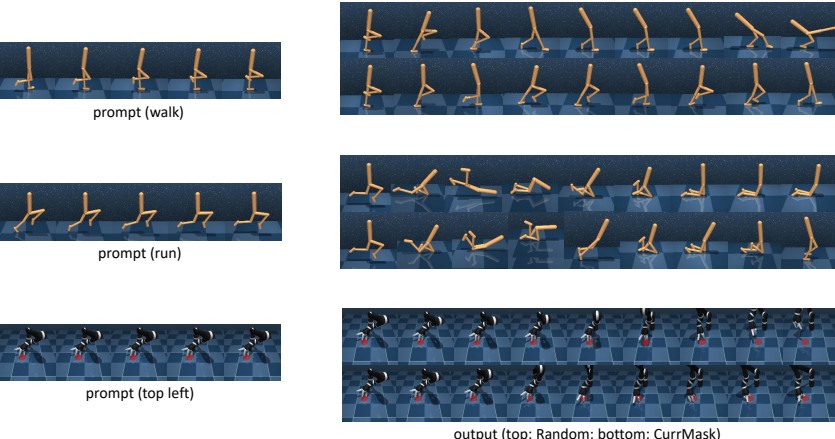

Figure 5: Skill prompting visualization.

# E ADDITIONAL EXPERIMENTAL RESULTS

## E.1 RESULTS ON HALFCHEETAH

In addition to the environments from the DeepMind control suite, we also conduct experiments on the HalfCheetah environment from the OpenAI Gym benchmark. We follow the setup described in Xu et al. (2022). In the HalfCheetah environment, the agent is tasked with achieving certain goal velocities. We consider three tasks with goal velocities $v_{\text{goal}}$ of 1.0 m/s, 2.0 m/s, and 3.0 m/s respectively. The reward is calculated by summing two parts and then normalizing the result to $[0, 1]$:

$$r_{\text{goal}} = -1.0 \times |v - v_{\text{goal}}|, \quad r_{\text{ctrl}} = -0.05 \times \|a\|^2.$$

We use the same protocol to collect pretraining data and pretrain the model as we do for the `Walker` and `Jaco` environments. However, when it comes to downstream tasks, we find that using unsupervised exploratory data for offline RL finetuning does not produce meaningful results. As a result, we instead subsample the validation data for each environment to construct a higher quality dataset for offline RL.

The results are shown in Table 5. It is evident that CurrMask outperforms `Random` across most tasks. For `Mixed-prog`, although it excels in goal-conditioned planning tasks, it performs worse than `Random` in both skill prompting and offline RL. We believe it is because the manually designed curriculum used by `Mixed-prog` forces the model to overfit on long-horizon prediction tasks at the end of training. In contrast, CurrMask possesses the ability to automatically balance these skills during training, highlighting its adaptivity and further confirming its advantages over other methods.

| skill ↑ | $v = 1.0$ | $v = 2.0$ | $v = 3.0$ |
|---|---|---|---|
| Random | $107.9_{\pm 0.2}$ | $95.1_{\pm 0.3}$ | $81.0_{\pm 0.3}$ |
| Mixed | $106.5_{\pm 0.2}$ | $92.6_{\pm 0.2}$ | $77.1_{\pm 0.4}$ |
| Mixed-prog | $102.9_{\pm 0.2}$ | $88.8_{\pm 0.2}$ | $69.1_{\pm 0.6}$ |
| CurrMask | $108.8_{\pm 0.1}$ | $97.3_{\pm 0.2}$ | $82.7_{\pm 0.4}$ |
| goal ↓ | $v = 1.0$ | $v = 2.0$ | $v = 3.0$ |
| Random | $4.34_{\pm 0.06}$ | $5.82_{\pm 0.13}$ | $7.78_{\pm 0.13}$ |
| Mixed | $4.52_{\pm 0.10}$ | $5.91_{\pm 0.11}$ | $7.68_{\pm 0.11}$ |
| Mixed-prog | $3.95_{\pm 0.08}$ | $4.98_{\pm 0.07}$ | $6.12_{\pm 0.07}$ |
| CurrMask | $4.49_{\pm 0.08}$ | $5.98_{\pm 0.10}$ | $7.60_{\pm 0.11}$ |
| offline RL ↑ | $v = 1.0$ | $v = 2.0$ | $v = 3.0$ |
| Random | $812.8_{\pm 12.9}$ | $759.6_{\pm 61.2}$ | $495.5_{\pm 77.5}$ |
| Mixed | $820.1_{\pm 17.3}$ | $725.7_{\pm 110.4}$ | $530.5_{\pm 88.2}$ |
| Mixed-prog | $817.9_{\pm 15.5}$ | $712.6_{\pm 72.1}$ | $490.9_{\pm 82.4}$ |
| CurrMask | $817.5_{\pm 7.7}$ | $776.6_{\pm 35.8}$ | $515.7_{\pm 72.0}$ |

Table 5: Halfcheetah results.

## E.2 IMPACT OF MASKING CURRICULA

In Figure 6, we plot the cumulative reward of each 30 steps of the generated trajectory. The patterns of `Mixed-prog` and CurrMask are substantially different. During the initial stage of pretraining, `Mixed-prog` (trained with small blocks only) struggles to learn skills at all levels of temporal granularity. CurrMask however exhibits faster skill acquisition and adapts its masking scheme dynamically during training.

Figure 7 illustrates the mean block size used by CurrMask while pretraining on `reach_top_left`, as a function of training steps. We notice an upward trend in block size, suggesting that CurrMask progressively enhances masking difficulty.

## E.3 RESULTS OF USING CONSTANT BLOCK SIZES

Table 6 compares CurrMask and `Mixed-`$k$, which employs a fixed block size $k$ but with uniformly sampled masked ratios. We can observe that the optimal value of $k$ for `Mixed-`$k$ varies depending on the downstream task, whereas the results of CurrMask are more consistent.

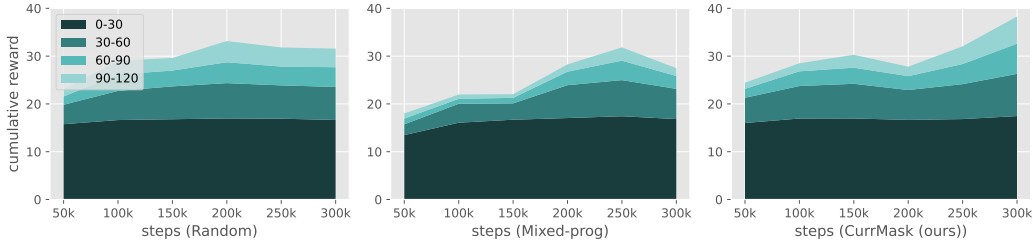

Figure 6: Skill prompting performance on Walker run in the pretraining phase.

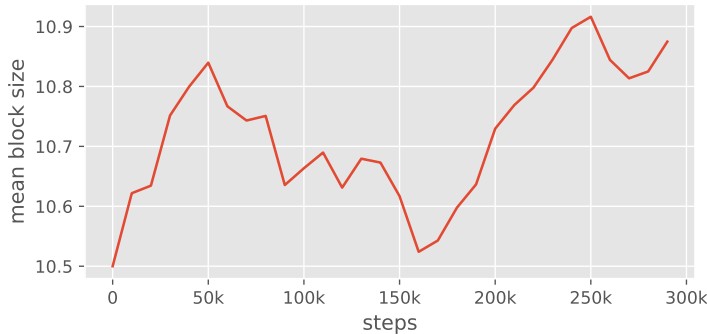

Figure 7: Mean block size vs. training steps during pretraining.

### E.4 ALTERNATIVE AUTOMATED CURRICULUM LEARNING METHODS

To strengthen our evaluation, we conduct experiments on another curriculum learning method (Mati-isen et al., 2019). We consider two variants that "are competitive in both and has least hyperparam-eters": Sampling which keeps a buffer of last $K$ rewards for each task and samples the new task using Thompson sampling. The two variants differ only in whether tasks are sampled according to the absolute reward values (Sampling-T) or not (Sampling-F). Table 7 shows that CurrMask achieves better performances in the considered tasks. This indicates that EXP3, with normalized rewards, better handles the non-stationarity of training dynamics.

| skill ↑ | reach-bl | reach-br | reach-tl | reach-tr | run | stand | walk |
|---|---|---|---|---|---|---|---|
| Mixed-1 | 64.3±5.5 | 63.1±4.2 | 62.4±4.3 | 67.4±3.9 | 30.4±0.9 | 105.1±3.4 | 58.7±4.0 |
| Mixed-5 | 31.4±2.7 | 29.9±2.4 | 32.6±2.4 | 34.9±1.4 | 14.1±0.3 | 97.4±2.6 | 29.4±1.5 |
| Mixed-10 | 63.5±6.6 | 62.0±2.4 | 65.0±4.3 | 72.1±4.5 | 18.2±0.6 | 106.2±4.8 | 34.8±1.5 |
| Mixed-15 | 75.2±6.6 | 72.4±4.7 | 72.4±4.0 | 75.1±4.6 | 33.0±1.0 | 110.3±3.1 | 80.5±1.5 |
| Mixed-20 | 79.4±6.4 | 77.6±4.5 | 79.0±4.9 | 83.4±3.6 | 33.5±1.4 | 111.8±2.9 | 73.3±4.0 |
| CurrMask | 81.7±7.4 | 84.5±2.9 | 78.7±6.1 | 83.5±3.7 | 37.1±1.4 | 109.7±3.3 | 90.6±2.4 |
| goal ↓ | reach-bl | reach-br | reach-tl | reach-tr | run | stand | walk |
| Mixed-1 | 1.50±.06 | 1.51±.04 | 1.45±.05 | 1.44±.06 | 15.45±.47 | 4.83±.44 | 10.11±.21 |
| Mixed-5 | 2.26±.08 | 2.28±.06 | 2.29±.08 | 2.36±.12 | 18.66±.58 | 5.66±.56 | 12.62±.33 |
| Mixed-10 | 1.61±.06 | 1.63±.04 | 1.59±.05 | 1.55±.08 | 15.80±.48 | 4.90±.44 | 10.51±.28 |
| Mixed-15 | 1.72±.05 | 1.80±.03 | 1.75±.06 | 1.77±.07 | 15.37±.61 | 4.82±.44 | 10.20±.25 |
| Mixed-20 | 1.51±.07 | 1.55±.04 | 1.49±.06 | 1.48±.07 | 15.56±.59 | 4.86±.44 | 10.17±.12 |
| CurrMask | 1.45±.05 | 1.47±.05 | 1.41±.06 | 1.38±.08 | 15.48±.49 | 4.80±.44 | 10.05±.20 |
| offline RL ↑ | reach-bl | reach-br | reach-tl | reach-tr | run | stand | walk |
| Mixed-1 | 151.8±48.8 | 170.4±45.4 | 152.7±57.4 | 166.9±23.5 | 159.1±49.0 | 684.7±188.5 | 440.0±226.0 |
| Mixed-5 | 175.2±25.7 | 169.9±45.7 | 177.8±14.2 | 193.6±19.7 | 212.8±52.9 | 866.7±78.7 | 316.3±118.8 |
| Mixed-10 | 126.3±101.0 | 196.4±35.0 | 187.7±24.8 | 181.5±68.0 | 183.9±15.2 | 847.7±100.1 | 321.9±124.5 |
| Mixed-15 | 177.9±66.3 | 193.2±11.2 | 163.6±23.1 | 167.9±30.0 | 122.2±79.7 | 722.9±68.5 | 447.4±144.2 |
| Mixed-20 | 193.8±15.0 | 207.4±16.9 | 178.4±38.2 | 191.0±11.7 | 212.4±55.8 | 829.9±78.2 | 437.8±253.1 |
| CurrMask | 186.0±12.9 | 158.9±55.0 | 199.6±20.0 | 154.4±48.4 | 258.5±45.5 | 843.3±84.6 | 610.9±76.9 |

Table 6: Comparison between CurrMask and Mixed-$k$.

| skill ↑ | reach-bl | reach-br | reach-tl | reach-tr | run | stand | walk |
|---|---|---|---|---|---|---|---|
| Sampling-F | 55.7±5.6 | 53.6±3.5 | 56.8±3.3 | 60.9±3.3 | 31.8±1.1 | 110.6±3.7 | 53.9±2.7 |
| Sampling-T | 59.9±5.0 | 58.6±3.4 | 62.2±3.8 | 67.2±3.3 | 25.6±0.5 | 109.3±4.1 | 52.6±2.0 |
| CurrMask | 81.7±7.4 | 84.5±2.9 | 78.7±6.1 | 83.5±3.7 | 37.1±1.4 | 109.7±3.3 | 90.6±2.4 |
| goal ↓ | reach-bl | reach-br | reach-tl | reach-tr | run | stand | walk |
| Sampling-F | 1.48±0.08 | 1.48±0.03 | 1.45±0.07 | 1.43±0.06 | 15.84±0.55 | 4.93±0.44 | 10.22±0.24 |
| Sampling-T | 1.50±0.08 | 1.57±0.04 | 1.46±0.06 | 1.47±0.09 | 15.94±0.52 | 4.90±0.45 | 10.21±0.22 |
| CurrMask | 1.45±0.05 | 1.47±0.05 | 1.41±0.06 | 1.38±0.08 | 15.48±0.49 | 4.80±0.44 | 10.05±0.20 |
| offline RL ↑ | reach-bl | reach-br | reach-tl | reach-tr | run | stand | walk |
| Sampling-F | 181.7±20.6 | 167.7±18.6 | 176.0±28.8 | 146.8±52.3 | 247.9±53.9 | 675.7±281.1 | 350.7±248.5 |
| Sampling-T | 175.8±8.3 | 188.4±17.3 | 174.4±23.5 | 164.7±19.8 | 207.6±31.7 | 781.0±88.8 | 574.0±249.4 |
| CurrMask | 186.0±12.9 | 158.9±55.0 | 199.6±20.0 | 154.4±48.4 | 258.5±45.5 | 843.3±84.6 | 610.9±76.9 |

Table 7: Results of automated curriculum learning methods in Matiisen et al. (2019).

## F   ATTENTION VISUALIZATION

In this section, we provide more details about our attention map visualization and additional results.

**Setup**   To provide a clearer visualization of the differences between CurrMask and other baselines in zero-shot skill prompting and goal reaching, we visualize the attention maps of their first layer decoders.For comparison purposes, we employ two masking techniques: prompt masking and goal masking. Prompt masking masks all tokens except the first 8 tokens, while goal masking masks all tokens except two randomly sampled state tokens.

Specifically, we evaluate the aforementioned masking methods on 10 trajectories randomly sampled from validation sets using the pretrained model. We then compute the average attention map for each technique, and finally apply L2 normalization to the attention maps of all four heads to obtain the first layer attention map. We focus on a truncated token sequence of length 32, resulting in a final attention map of size $32 \times 32$ for clearly demonstrating the differences.

**Additional Results**   We provide additional visualization results in Figure 8-9. Apart from the observation that CurrMask better captures long-term dependencies than `Random`, we find that increasing the block size for `Block-wise` leads to greater capabilities in long-term prediction, which supports our intuition regarding the benefits brought by block-wise masking.

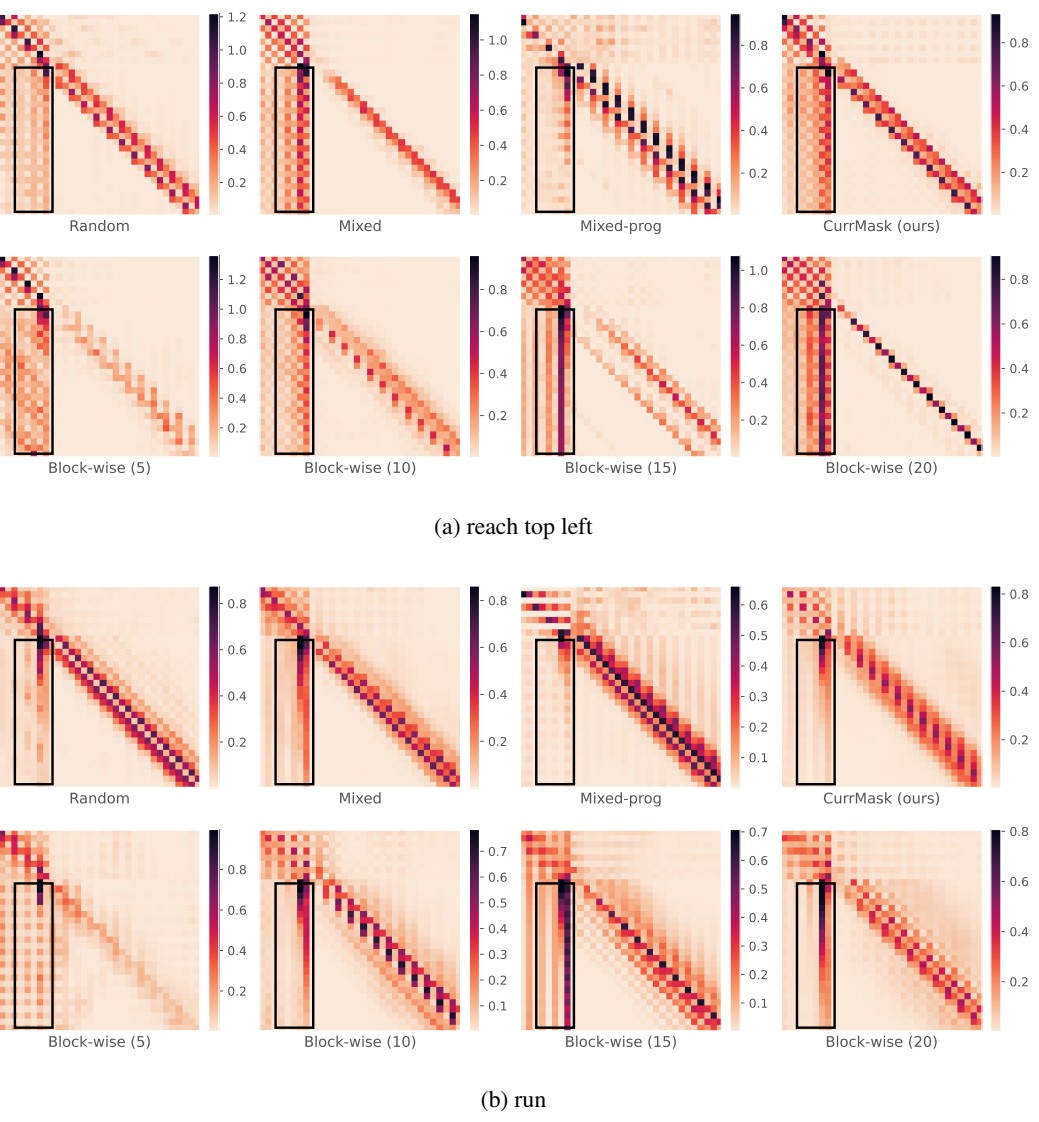

(a) reach top left

(b) run

Figure 8: Attention visualization with prompt masking.

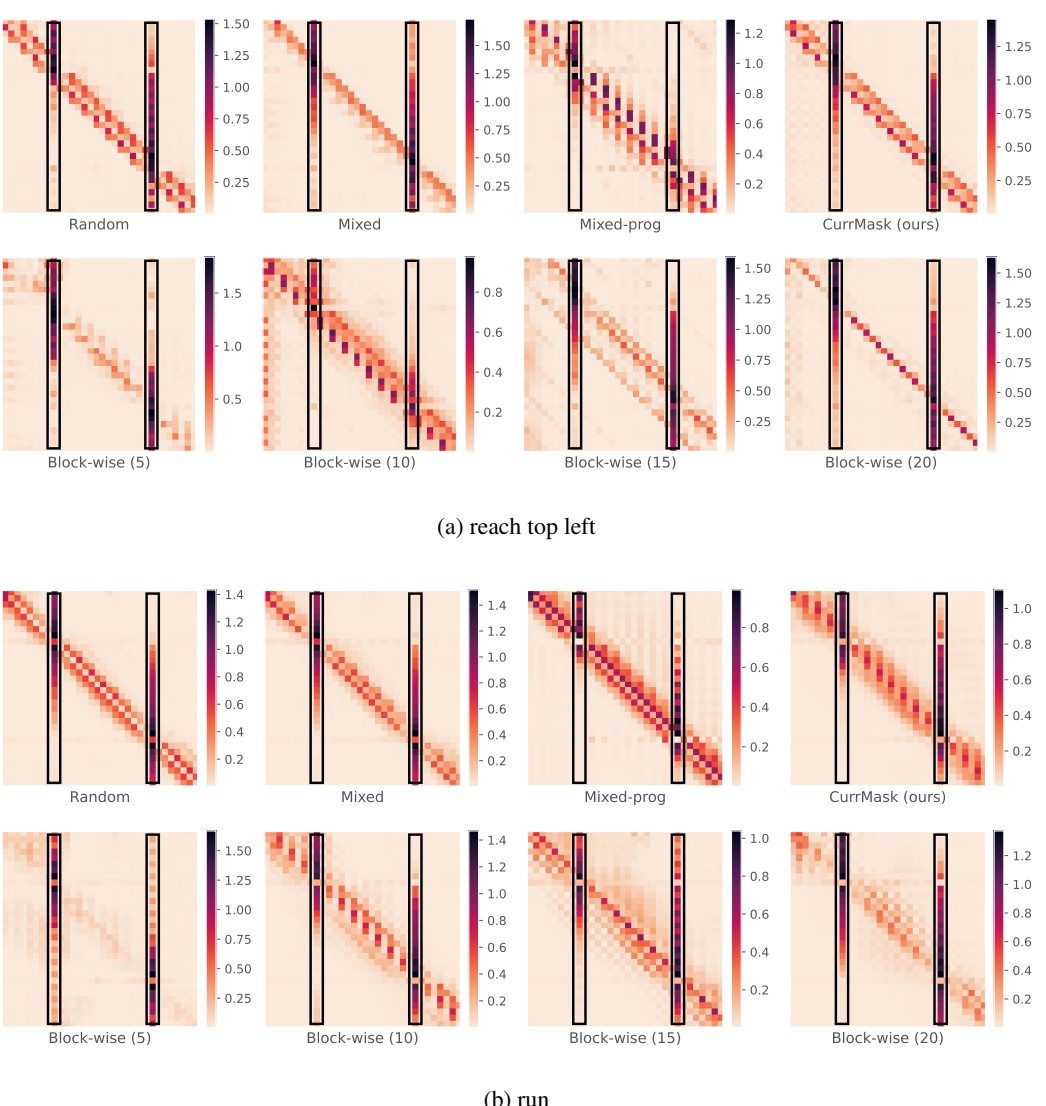

(a) reach top left

(b) run

Figure 9: Attention visualization with goal masking.

