# OpenReview forum: "CurrMask: Learning Versatile Skills with Automatic Masking Curricula"
_ICLR.cc/2024/Conference — ICLR 2024 Conference Withdrawn Submission_

### Official Review · Reviewer_hUSq · 2023-10-30

**Soundness:** 2 fair
**Presentation:** 2 fair
**Contribution:** 3 good
**Rating:** 5
**Confidence:** 4

**Summary:**

This paper introduces a curriculum masking approach that can adjust the masking scheme. To implement this approach, this paper leverages a multi-armed bandit model to adjust its mask based on the learning progress during training. A series of experiments are conducted on the DeepMind control suite to show the performance of CurrMask. This paper has a clear and logical structure, and the experiments are relatively rational.

**Strengths:**

1) It makes sense to use block masking with a mixed mask ratio and block size to capture temporal dependencies at both short-term and long-term levels.
2) It is reasonable and easy to model the finding of a suitable curriculum of study for masking schemes as a multi-armed bandit.

**Weaknesses:**

1) The comparisons between CurrMask and similar methods or state-of-the-art baselines are insufficient.
2) It needs to give more explanations and details to subsection “Evaluation of Long-term Prediction”, as information obtained from this set of experiments and Figure 4 is not enough.
3) The role of CurrMask for learning useful and generic representations for downstream finetuning and accelerating offline reinforcement learning are not well explained and fully experimentally validated.

**Questions:**

1) Is there a reason to use different random seeds to construct the validation set used for zero-shot evaluation? Random seeds in devices cannot replace changes in the environment, which is also different from generic zero-shot generalization setups that use background video or color changes.
2) In Table 1, Currmask performs worse than Mixed-prog despite the fact that all methods using block prediction are better than Random. Does this suggest that the proposed selection of appropriate curriculum is not better than manual design (progressively increases the block size)?
3) In Table 2, Mixed has a worse performance than Random. Does this mean that block-wise masking is not conducive to long-term planning?

---

### Official Review · Reviewer_nmaR · 2023-10-30

**Soundness:** 2 fair
**Presentation:** 3 good
**Contribution:** 2 fair
**Rating:** 5
**Confidence:** 4

**Summary:**

This paper proposes a learnable masking strategy for pre-training representations. More specifically, it uses a multi-armed bandit algorithm to learn the masking ratio and block size. Since it assumes no task information at the time of pre-training, the paper uses target loss decrease on the offline dataset as a proxy for the rate of solving downstream tasks.

It experiments with a set of environments in the DeepMind Control Suite. On zero-shot skill prompting and goal-conditioned planning, it shows CurrMask is better than random masking and progressive block masking. On offline RL, it shows CurrMask can achieve higher cumulative rewards though sometimes falling behind in the middle of training.

The ablations on different mask block sizes and ratios show the model prefers larger blocks while maintaining a medium mask ratio as training progresses.

**Strengths:**

Studies an interesting and important problem: how to pre-train representations for control tasks and proposes a solution that learns the mask during training.

The method is well-motivated from the observation that different pre-defined fixed masking strategies perform differently in different tasks and there could be a learnable way to combine the best of all masking techniques.

The proposed algorithm is simple and paper writing is easy to follow.

**Weaknesses:**

In offline RL results, it seems CurrMask sometimes is worse in the middle of training. Why is that?

In the Jaco environment, CurrMask is worse than progressive masking on skill prompting, but better on goal-conditioned planning. Why isn’t CurrMask universally better as it could learn any combination of mask strategies theoretically? Any ideas on the set of tasks that CurrMask should be better at?

As pointed out in the limitations, CurrMask adds overhead to training as it requires a bandit model for scheduling. A comparison of training wall-clock time would be nice.

**Questions:**

Why is CurrMask sometimes worse in the middle of offline RL training?

Why is CurrMask sometimes worse in skill prompting?

Could you show a comparison of training wall-clock time between CurrMask and other fixed masking strategies?

---

### Official Review · Reviewer_zZrj · 2023-10-30

**Soundness:** 2 fair
**Presentation:** 2 fair
**Contribution:** 2 fair
**Rating:** 5
**Confidence:** 3

**Summary:**

The paper presents a method that learns skills through curriculum masking. Specifically, the approach CurrMask can automatically arranges the order of different masking schemes for training. The algorithm is tested on Deepmind Control Suite tasks, and show positive results in representation learning, zero-shot skill prompting, and zero-shot goal-conditioned planning.

**Strengths:**

1. The idea of designing different masking curriculum to learn different skills is generally interesting and makes sense.
2. Code is provided in the supplementary material and hyperparameters used in the experiments are provided in the appendix.

**Weaknesses:**

1. There is not enough explanation for the proposed "task selection" method, which I believe is the central part of the proposed approach. Specifically, in section 4.3, what do \omega_i, \omega'_i, K denote? Does \pi represent the policy? What is the intuition behind the two equations in the task selection subsection? Without explaining these, it is hard for me to understand how the proposed approach learn the masking curriculum.

2. There is no visualization of the proposed masking curriculums. As this is the central contribution of the paper, it would be very interesting to see what the actual masking curriculum is for those continuous control tasks and how is affect the numerical results.

3. From table 1 and table 2, it seems the proposed approach CurrMask is only slightly better the baseline masking schemes in a few scenarios (<50%).

4. There is not comparison to other existing skill learning methods with offline data, e.g. the two papers (Ajay et al., 2021; Jiang et al., 2022) mentioned in the paper's related work section.

**Questions:**

See weaknesses.

---

### Official Review · Reviewer_RhwK · 2023-11-04

**Soundness:** 3 good
**Presentation:** 3 good
**Contribution:** 2 fair
**Rating:** 3
**Confidence:** 4

**Summary:**

This paper proposed a curriculum block-wise masking approach that can adjust the masking scheme automatically based on the learning progress, and demonstrate it's performance on MuJoCo continuous control tasks over other masking strategy.

**Strengths:**

1. In general, the writing is clear and easy to follow.
2. The proposed block-wise masking strategy and masking scheme curricula showed visible advancements in skill prompting tasks in both the Walker and Jaco domains.
3. The visualization showed that the proposed approach learned better long-term dependencies.

**Weaknesses:**

1. Limited improvements on goal-reaching and RL

    From Table. 1, 2, and Figure 2, the benefits of the proposed approach are most significant on skill prompting, but not goal reaching and offline RL. It's also understandable that these two tasks may not require long-term dependencies. The results will be stronger and more convincing if this approach is tested on reaching long-horizon goals, or multiple future goals spanning in long term.


2. Limited domains and Tasks

    This paper only studies tasks in Walker, Cheetah, and Jaco, which are relatively low-dimensional. It could be more convincing with more investigation on some other high-dimensional control or manipulation domains.

3. Limited baselines

    This paper compared the advantages of masking schemes with different mask-based transformer-based approaches, which are more like self-ablation studies. It would be more convincing to compare with other autoregressive variants (like goal-conditioned / skill-conditioned DT) in these tasks and domains.

**Questions:**

1. Figure 3(a) addressed the importance of block-wise masking on skill prompting tasks. For the other two tasks, will the block-wise masking have the same effects?

2. For Mixed-Prog, the description is confusing: "Within the intervals of [[0, 0.25), [0.25, 0.5), [0.5, 0.75), [0.75, 1.00)], sub-sequences are sampled with lengths ranging from [[1, 5], [1, 10], [1, 15], [1, 20]]". Can you explain why you design Mixed-Prog in this way? Why not choose the set of fixed numbers as in CurrMask?

3. "A unified model trained on diverse multi-task data..." is mentioned only in the skill prompting paragraph. Does this mean for your other two tasks, the model is not trained on a mixed multi-task dataset?

---

### Author Response · Authors · 2023-11-22

Thank you for the reviewers' insightful and comprehensive feedback. After careful consideration, we have decided to withdraw our paper.